# Recursive SNE: Fast Prototype-Based t-SNE for Large-Scale and Online Data

**Agil Aghasanli**[*]                                     *a.aghasanli1@lancaster.ac.uk*
*Lancaster Intelligent, Robotic and Autonomous systems (LIRA) Centre*
*School of Computing and Communications*
*Lancaster University, UK*

**Plamen Angelov**[*]                                     *p.angelov@lancaster.ac.uk*
*Lancaster Intelligent, Robotic and Autonomous systems (LIRA) Centre*
*School of Computing and Communications*
*Lancaster University, UK*

**Reviewed on OpenReview:** *https://openreview.net/forum?id=7wCPAFMDWM*

## Abstract

Dimensionality reduction techniques like t-SNE excel at visualizing structure in high-dimensional data but incur high computational costs that limit their use on large or streaming datasets. We introduce the Recursive SNE (RSNE) framework, which extends t-SNE with two complementary strategies: i-RSNE for real-time, point-wise updates and Bi-RSNE for efficient batch processing. Across diverse settings, including standard image benchmarks (CIFAR10/CIFAR100) with DINOv2 and CLIP features, domain-specific iROADS road scenes, neuroimaging data from the Haxby fMRI dataset, and long-term climate records, RSNE delivers substantial speedups over Barnes–Hut t-SNE while maintaining or even improving cluster separability. By combining a lightweight prototype-based initialization with localized KL-divergence refinements, RSNE offers a scalable and adaptable framework for both large-scale offline embedding and on-the-fly visualization of streaming data.

## 1 Introduction

In the big data era, high-dimensional datasets are standard in many fields, including finance (Fan et al., 2011), image processing, and biology (Wong, 2016). The "curse of dimensionality" (Bellman, 1957; Friedman, 1997), which may conceal underlying patterns and structures, makes it very difficult to analyse and visualise such data. Dimensionality reduction methods have emerged as essential tools for addressing these challenges by converting high-dimensional data into more manageable formats while maintaining crucial information. For example, in genomics, these techniques help identify significant gene expression patterns (Golub et al., 1999); in natural language processing, they help reduce the complexity of text representations (Pennington et al., 2014); in computer vision, they facilitate the extraction of meaningful features from images (Turk & Pentland, 1991).

Several methods have been developed to address the challenges of high-dimensional data. Principal Component Analysis (PCA) (Pearson, 1901) is a widely used technique that transforms data into a set of orthogonal components, capturing the maximum variance in the data. Linear Discriminant Analysis (LDA) (Fisher, 1936) focuses on finding linear combinations of features that best separate different classes in the data space. The nonlinear method known as t-Distributed Stochastic Neighbour Embedding (t-SNE) (van der Maaten & Hinton, 2008) attempts to maintain the probability distribution of pairwise similarities by modeling each high-dimensional data by a two- or three-dimensional position. Uniform Manifold Approximation and Projection (UMAP) (McInnes & Healy, 2018) emphasizes both local and global data structure, providing faster

---

[*]Equal contribution.

computation and often better preservation of global structures compared to t-SNE. Isomap (Tenenbaum et al., 2000) extends Multidimensional Scaling (Torgerson, 1952) by incorporating geodesic distances, capturing the intrinsic geometry of the data manifold. Locally Linear Embedding (LLE) (Roweis & Saul, 2000) seeks to preserve local relationships by reconstructing each data point from its nearest neighbors.

Despite advances in dimensionality reduction techniques, many methods face challenges when dealing with streaming or large-scale datasets. Traditional t-SNE, for example, requires reprocessing the entire dataset when new data points are added, making it computationally expensive for real-time applications. To address this limitation, we propose Recursive SNE (RSNE), an extension of t-SNE that allows for efficient updates with new data points without necessitating a complete recomputation. Specifically, *i-RSNE* performs fast, point-wise updates ideal for streaming data, while *Bi-RSNE* handles large offline batches with vectorized operations. This approach is particularly beneficial for applications that involve dynamic or streaming data, where timely visualization and analysis are crucial.

In our experiments, Bi-RSNE processes transformer-derived features from CIFAR10 (Krizhevsky et al., 2009) and CIFAR100 (Krizhevsky et al., 2009) approximately five times faster than Barnes–Hut t-SNE (van der Maaten, 2013), while maintaining or enhancing silhouette (Rousseeuw, 1987) and Davies–Bouldin (Davies & Bouldin, 1979) scores across both DINOv2 (Oquab et al., 2023) and CLIP (Radford et al., 2021) embeddings. For instance, the batch method Bi-RSNE achieved comparable cluster compactness within roughly 40 % of the original computation time on CIFAR100. Meanwhile, i-RSNE integrates each new CIFAR sample in just a few milliseconds, over an order of magnitude faster than rerunning a full embedding, yet delivers virtually identical cluster layouts after a single gradient step. On the domain-specific iROADS dataset, Bi-RSNE achieves around 3 times speedup and yields sharper, more semantically coherent clusters. On neuroimaging data from the Haxby fMRI dataset (Haxby et al., 2001), RSNE achieves up to a 2.5× speedup over Barnes–Hut t-SNE while preserving embedding quality, making it suitable for interactive exploration of brain activity patterns. Finally, applied to a ten-year climate stream (Muthu, 2017), RSNE reduces runtime by nearly 50 % while preserving cluster separation, underscoring its effectiveness for both offline and real-time embeddings.

In this paper, we introduce RSNE and demonstrate its effectiveness in handling high-dimensional data in dynamic environments. Our contributions include:

- Development of the RSNE framework, including i-RSNE for incremental updates and Bi-RSNE for batch updates, reducing computational overhead.

- Comprehensive evaluation of RSNE's performance compared to Barnes–Hut t-SNE, highlighting its advantages in terms of efficiency and scalability.

- Application of RSNE to real-world datasets, including CIFAR10, CIFAR100, iROADS, and climate data, to demonstrate its flexibility and practical utility.

## 2 Related Work

### 2.1 Overview of Dimensionality Reduction

Dimensionality reduction techniques are broadly categorized into linear and nonlinear methods (Cunningham & Ghahramani, 2015). Linear methods, such as PCA (Pearson, 1901) and LDA (Fisher, 1936), assume that the data lies on or near a linear subspace of the high-dimensional space. These techniques aim to find linear projections that capture the most significant variance (PCA) or that best separate different classes (LDA). On the other hand, nonlinear methods are designed to capture more complex structures in the data that cannot be adequately represented by linear projections. Techniques like t-SNE (van der Maaten & Hinton, 2008), UMAP (McInnes & Healy, 2018), Isomap (Tenenbaum et al., 2000), and LLE (Roweis & Saul, 2000) fall into this category, as they aim to preserve the intrinsic geometry of the data manifold by considering nonlinear relationships between data points.

### 2.1.1 t-SNE and Its Limitations

t-Distributed Stochastic Neighbor Embedding (t-SNE) has become a popular tool for visualizing high-dimensional data due to its ability to preserve local structures and reveal clusters. However, t-SNE has notable limitations, particularly concerning scalability and the handling of new data points. The algorithm's computational complexity is $\mathcal{O}(N^2)$ due to the pairwise similarity calculations, making it challenging to apply to large datasets. Additionally, it requires reprocessing the entire dataset when new data points are added, which is impractical for streaming data applications.

### 2.1.2 Barnes-Hut t-SNE

To address the scalability issue, Barnes-Hut t-SNE (van der Maaten, 2013) was developed, which speeds up computations using a quadtree-based approximation. This approach reduces the computational complexity to $\mathcal{O}(N \log N)$, making it more feasible to apply t-SNE to larger datasets. However, Barnes-Hut t-SNE still requires reprocessing all data when new points are added, limiting its applicability in dynamic or streaming data scenarios.

### 2.1.3 Parametric t-SNE and FIt-SNE

Other techniques have been proposed to handle large-scale data and dynamic updates. Parametric t-SNE (van der Maaten, 2009) learns a mapping function via deep learning, allowing for the embedding of new data points without retraining on the entire dataset. However, this approach is limited by training complexity and may not capture the intricate structures in the data as effectively as non-parametric methods. FIt-SNE (Fourier-Interpolated t-SNE) (Linderman et al., 2019) uses fast matrix operations to improve computational efficiency but does not support streaming updates, as it still relies on batch processing of data.

## 2.2 Data Visualization at Scale

Large datasets require visualization systems that can process and render millions of points without sacrificing interactivity or clarity.

Scalable frameworks typically preprocess data into multi-resolution hierarchies or employ on-the-fly aggregation so that only a subset of the data is rendered at any zoom level (Childs et al., 2011). GPU-accelerated pipelines and parallel rendering engines further ensure that panning, zooming, and filtering remain fluid even as dataset size grows. Server–client architectures often couple a lightweight front end with a server-side data reduction layer, dynamically streaming only the detail needed for the current view.

Graph-based methods such as LargeVis construct an approximate $k$-nearest-neighbor graph once and then optimize a force-directed layout via edge-centric stochastic updates (Tang et al., 2016). By decoupling neighbor search (e.g. using locality-sensitive hashing) from layout optimization, these approaches embed millions of nodes in minutes rather than hours. Asynchronous graph partitioning and parallel force computations further improve scalability on commodity hardware.

RSNE extends these ideas by performing only local, constant-time updates for incoming points, assigning them into existing clusters and refining their positions with a fixed number of gradient steps. This enables truly incremental graph expansions without re-running the full layout.

## 2.3 Manifold Learning for Temporal or Evolving Data

Many real-world streams, such as sensor networks, neuroimaging, climate records, change continuously, requiring embeddings that evolve smoothly rather than abrupt shifts at each time step.

In sliding-window fMRI analyses, functional connectivity patterns drift along a low-dimensional manifold as cognitive states change (Hutchison et al., 2013). Traditional approaches recompute embeddings on each window or interpolate between static layouts, often resulting in abrupt visual discontinuities. Some methods introduce temporal regularization into the embedding cost, but still require expensive batch computations.

Streaming manifold techniques update neighborhood graphs and eigenmaps incrementally, e.g., Streaming Isomap (Mahapatra & Chandola, 2017), to reflect evolving topology, but typically incur $\mathcal{O}(n \log n)$ or higher per update. To improve speed, these methods sometimes prune old points or coarsen graphs, sacrificing long-range structure for performance.

RSNE maintains cluster summaries (means and variances) and applies a single KL-divergence gradient step per new point or batch, achieving constant-time updates that preserve both local neighborhoods and global manifold shape.

## 2.4 Visualization in Domain-Specific Applications

Different application domains impose unique constraints on embedding quality and update speed. Geospatial analysis requires accurate spatial relationships, while biological imaging requires the preservation of complex high-dimensional patterns.

Geovisualization systems layer thematic data traffic flows, pollution levels, demographic indicators on geographic maps, demanding that the embedded proximity reflect real-world distances (Tominski & Schumann, 2020). Real-time sensor streams must be integrated without distorting existing clusters or misleading users, requiring incremental update strategies.

High-content microscopy produces feature vectors in the hundreds or thousands of dimensions, which are projected for tasks such as cell-type discovery or anomaly detection (Peng et al., 2010). When new image batches arrive (e.g., additional tissue sections or time-lapse frames), full re-embedding disrupts interpretability and delays analysis.

RSNE may let domain experts immediately visualize where they fall relative to existing clusters (see Section 4), keeping the analytic pipelines responsive while preserving the semantic cluster structure.

## 2.5 Transformer-Based Foundation Models

Transformer architectures have become foundational for generating rich, high-dimensional representations in both vision and language domains. In computer vision, the Vision Transformer (ViT) treats image patches as tokens in a pure transformer model, achieving state-of-the-art performance on large-scale image classification benchmarks (Dosovitskiy et al., 2020). Building on this idea, self-supervised variants such as DINOv2 learn visual features without labels by aligning patch embeddings across views, resulting in semantically meaningful clusters in feature space (Oquab et al., 2023). CLIP extends the transformer paradigm to a joint image–text setting, utilizing large-scale contrastive pretraining to align visual concepts with textual descriptions, thereby enabling zero-shot generalization (Radford et al., 2021).

On the language side, transformer models like BERT generate contextualized token embeddings that encode both syntactic and semantic information, markedly improving tasks such as question answering and named-entity recognition (Devlin et al., 2019). Autoregressive transformers, such as GPT, demonstrate few- and zero-shot capabilities by training on massive text corpora, supporting diverse downstream applications with minimal or no fine-tuning (Brown et al., 2020). Because transformer-based embeddings often form well-separated clusters, which reflect object classes in vision or thematic groupings in text, many visualization pipelines now operate directly on these pre-trained features. RSNE leverages this structure by initializing a small set of prototypes drawn from a transformer embedding space and then refining locally via KL-divergence steps. This ensures that the global layout induced by DINOv2 or CLIP remains intact while avoiding the need to rerun full t-SNE when new batches or streaming data arrive.

## 2.6 Online and Recursive Update Methods

Many online learning and outlier-detection methods rely on recursive updates to avoid storing all past observations. One prominent example is Recursive Density Estimation (RDE) (Angelov, 2012). RDE maintains a running mean and scalar summary (e.g., the sum of squared norms) so that the density estimate can be updated in current time instant from these summaries and the current sample. This exact,

nonparametric formulation supports real-time novelty detection and clustering without retaining the entire data history.

Similar ideas are also present in streaming clustering and dimensionality-reduction algorithms. Streaming K-means updates each centroid incrementally by combining the previous centroid with the new observation, avoiding full batch recomputation (Bottou & Bengio, 1994). Incremental principal component analysis (IPCA) maintains running covariance estimates via recursive formulas, adjusting principal components as new data arrive without re-computing eigenvectors from scratch (Weng et al., 2003).

RSNE integrates these recursive estimation principles directly into t-SNE. In the *i-RSNE* variant, each new data point is first assigned to its nearest high-dimensional prototype; that prototype's mean and variance are updated incrementally, precisely as in RDE, ensuring constant-time updates. The corresponding low-dimensional centroid is also updated recursively before a single KL-divergence gradient step adjusts the new point's embedding. In the *Bi-RSNE* variant, prototype updates and affinity computations are vectorized across a mini-batch, yet still rely on the same recursive summaries to maintain up-to-date prototypes. As a result, RSNE achieves efficient streaming and batch updates, which preserve both local neighborhoods and global layout without ever rerunning full t-SNE on the entire dataset.

## 3 Methodology

Throughout this paper, we denote the original datapoint in the input space by $x \in \mathbb{R}^D$, where $D$ is the dimensionality of the original data. The corresponding low-dimensional embedding is denoted by $y \in \mathbb{R}^d$, where $d \ll D$ is the dimension of the embedding space. We use $X = \{x_1, x_2, \ldots, x_N\}$ for the dataset of $N$ samples in the original space, and $Y = \{y_1, y_2, \ldots, y_N\}$ for their embeddings. Distances are Euclidean in both spaces.

### 3.1 i-RSNE

i-RSNE extends the initialization followed by updating paradigm to a fully online setting, embedding one new point at a time while preserving the existing layout.

#### 3.1.1 Initialization Phase

We first extract a small seed subset $X_{\text{init}} \subset X$ of size $\alpha N$, where $N$ is the total number of data points (e.g. $\alpha = 0.2$). On this subset, Barnes–Hut t-SNE produces low-dimensional coordinates $\{y_i^{(0)}\}$ that capture the global structure of the data. In parallel, K-means clustering in the original feature space yields high-dimensional centroids $\mu_k^{(0)}$ and variances $\sigma_k^{2\,(0)}$. We then compute low-dimensional centroids

$$\mu_{y,k}^{(0)} = \frac{1}{|\mathcal{C}_k|} \sum_{i \in \mathcal{C}_k} y_i^{(0)},$$

where $\mathcal{C}_k$ denotes the set of seed points assigned to cluster $k$. These paired high- and low-dimensional prototypes serve as the fixed foundation for all subsequent updates, ensuring that the coarse manifold layout remains stable as new data points arrive.

#### 3.1.2 Incremental Phase

When a new sample $x_{\text{new}}$ arrives, we first assign it to its nearest high-dimensional prototype:

$$k^* = \arg \min_k \|x_{\text{new}} - \mu_k^{(t-1)}\|_2.$$

This prototype search identifies the semantic region of the data manifold into which the new point most naturally falls. We then warm-start the embedding by placing the new point near the corresponding low-dimensional centroid:

$$y_{\text{new}}^{(t,0)} = \mu_{y,k^*}^{(t-1)} + \epsilon, \qquad \epsilon \sim \mathcal{N}(0, 0.1).$$

This initialization prevents the new point from overlapping existing points and provides a reasonable starting position for fine-tuning.

To keep our prototypes up to date without re-running K-means, we increment the cluster count $n_{k^*}$ by one and update its high-dimensional mean and second moment in constant time:

$$\mu_{k^*}^{(t)} = \frac{n_{k^*}^{(t-1)}}{n_{k^*}^{(t)}} \mu_{k^*}^{(t-1)} + \frac{1}{n_{k^*}^{(t)}} x_{\text{new}}, \quad \Sigma_{k^*}^{(t)} = \frac{n_{k^*}^{(t-1)}}{n_{k^*}^{(t)}} \Sigma_{k^*}^{(t-1)} + \frac{1}{n_{k^*}^{(t)}} \|x_{\text{new}}\|^2.$$

The new high-dimensional variance is then calculated as:

$$\sigma_{k^*}^{2\,(t)} = \Sigma_{k^*}^{(t)} - \left\| \mu_{k^*}^{(t)} \right\|^2.$$

To refine the new embedding, we compute high-dimensional affinities

$$\tilde{P}_k = \exp\!\left(-\|x_{\text{new}} - \mu_k^{(t)}\|^2 / (2\,\sigma_k^{2\,(t)})\right), \quad P_k = \frac{\tilde{P}_k}{\sum_m \tilde{P}_m},$$

and low-dimensional affinities

$$\tilde{Q}_k = \left(1 + \|y_{\text{new}}^{(t,0)} - \mu_{y,k}^{(t-1)}\|^2\right)^{-1}, \quad Q_k = \frac{\tilde{Q}_k}{\sum_m \tilde{Q}_m}.$$

These distributions quantify how strongly the new point prefers each prototype in both spaces. We then compute the gradient of the Kullback–Leibler divergence,

$$\nabla_y KL = 4 \sum_{k=1}^{K} (P_k - Q_k) \frac{y_{\text{new}}^{(t,0)} - \mu_{y,k}^{(t-1)}}{1 + \|y_{\text{new}}^{(t,0)} - \mu_{y,k}^{(t-1)}\|^2},$$

and take a single gradient step,

$$y_{\text{new}}^{(t,*)} = y_{\text{new}}^{(t,0)} - \eta\, \nabla_y KL.$$

This one–step refinement corrects the new point's position to better align local neighborhoods. The gradient descent update only affects the embedding of the new point, while the embeddings of previous points remain fixed. By restricting to a single iteration, we keep each update at $O(K\,(D+d))$ cost, where $d$ is the low-dimensional embedding dimension, enabling real-time streaming performance.

Once the final embedding $y_{\text{new}}^{(t,*)}$ is obtained, we update the low-dimensional centroid $\mu_{y,k^*}$ for cluster $k^*$ in constant time:

$$\mu_{y,k^*}^{(t)} = \frac{n_{k^*}^{(t-1)}}{n_{k^*}^{(t)}} \mu_{y,k^*}^{(t-1)} + \frac{1}{n_{k^*}^{(t)}} y_{\text{new}}^{(t,*)}.$$

These constant-time updates in the low-dimensional space ensure that our centroids accurately track the evolving embedding without requiring a full recomputation of cluster means.

### 3.2 Bi-RSNE

Bi-RSNE shares the same seed-set initialization as i-RSNE but replaces sequential, point-wise updates with a fully vectorized batch procedure in the incremental phase.

#### 3.2.1 Initialization Phase

Identical to i-RSNE, we extract a seed subset $X_{\text{init}}$, embed it with Barnes–Hut t-SNE to produce $\{y_i^{(0)}\}$, and cluster the seed by K-means to obtain high-dimensional prototypes $\{\mu_k^{(0)},\ \sigma_k^{2\,(0)}\}$ and low-dimensional centroids $\{\mu_{y,k}^{(0)}\}$. This shared initialization ensures both methods start from the same global layout.

### 3.2.2 Batch Update Phase

For a batch of $B$ new points $\{x_i\}_{i=1}^{B}$, we first compute the full $B \times K$ distance matrix $\|x_i - \mu_k^{(t-1)}\|^2$. The minimal entry in each row assigns $x_i$ to prototype $k_i$, and we collect these assignments into index sets $\mathcal{I}_k = \{i : k_i = k\}$. In one vectorized operation we update each high-dimensional centroid by

$$\mu_k^{(t)} = \frac{n_k^{(t-1)}}{n_k^{(t)}}\, \mu_k^{(t-1)} + \frac{1}{n_k^{(t)}} \sum_{i \in \mathcal{I}_k} x_i, \quad \Sigma_k^{(t)} = \frac{n_k^{(t-1)}}{n_k^{(t)}}\, \Sigma_k^{(t-1)} + \frac{1}{n_k^{(t)}} \sum_{i \in \mathcal{I}_k} \|x_i\|^2,$$

where $n_k^{(t)} = n_k^{(t-1)} + |\mathcal{I}_k|$.

Next, we compute unnormalized affinities for all $i, k$:

$$\tilde{P}_{i,k} = \exp\big(-\|x_i - \mu_k^{(t)}\|^2/(2\,\sigma_k^{2\,(t)})\big), \quad \tilde{Q}_{i,k} = \big(1 + \|y_i^{(t,u)} - \mu_{y,k}^{(t-1)}\|^2\big)^{-1},$$

normalize each row to obtain $P_{i,k}$ and $Q_{i,k}$, and compute the KL gradient for each point:

$$\nabla_{y_i} KL = 4 \sum_{k=1}^{K} (P_{i,k} - Q_{i,k}) \frac{y_i^{(t,u)} - \mu_{y,k}^{(t-1)}}{1 + \|y_i^{(t,u)} - \mu_{y,k}^{(t-1)}\|^2}.$$

A single vectorized gradient step

$$y_i^{(t,*)} = y_i^{(t,0)} - \eta\, \nabla_{y_i} KL$$

updates all $B$ embeddings simultaneously. By expressing distances, prototype updates, affinities, and gradients as batched matrix operations, Bi-RSNE achieves $O(B\,K\,(D + d))$ runtime, fully leveraging hardware parallelism to handle large-scale batches efficiently, where $B$ is the batch size, $D$ is the dimensionality of the original data, and $d$ is the low-dimensional embedding dimension.

Once the new low-dimensional embeddings $\{y_i^{(t,*)}\}_{i=1}^{B}$ are computed, we update each low-dimensional prototype $\mu_{y,k}$ in constant time for cluster $k$:

$$\mu_{y,k}^{(t)} = \frac{n_k^{(t-1)}}{n_k^{(t)}}\, \mu_{y,k}^{(t-1)} + \frac{1}{n_k^{(t)}} \sum_{i \in \mathcal{I}_k} y_i^{(t,*)}.$$

*Complexity.* Both i-RSNE and Bi-RSNE have total cost

$$O(\underbrace{nKDI}_{\text{seed clustering}}) + O(\underbrace{n \log n}_{\text{t-SNE init}}) + O(\underbrace{mK(D + d)}_{\text{incremental updates}}),$$

where $n$ is the seed size, $I$ the number of $k$-means iterations, and $m = N - n$ the number of incremental points. The difference lies in how the last term is executed: i-RSNE processes one point at a time, repeating the $K(D + d)$ work sequentially, whereas Bi-RSNE groups points into batches of size $B$ and performs the same operations in parallel, exploiting vectorization and hardware acceleration. This makes Bi-RSNE practically faster, even though the asymptotic orders are identical.

## 4 Experiments

### 4.1 Datasets

**CIFAR10 Dataset.** The CIFAR10 dataset (Krizhevsky et al., 2009) consists of 60,000 natural images in 10 classes (6,000 images per class), with 50,000 images for training and 10,000 for testing. We extract 1024-dimensional feature vectors from each image using pre-trained DINOv2 ViT-L/14 and CLIP ViT-L/14 models. These transformer-based embeddings capture rich semantic and visual information, providing challenging high-dimensional inputs. We evaluate both i-RSNE and Bi-RSNE on the train set of 50,000 images, comparing embedding quality (silhouette score, Davies–Bouldin index) and runtime against Barnes–Hut t-SNE.

---

**Algorithm 1** i-RSNE

---

**Require:** High-dimensional dataset $X$, seed fraction $\alpha$, number of clusters $K$, learning rate $\eta$
1: Select a random seed subset $X_{\text{init}} \subset X$ of size $\alpha|X|$
2: Compute high-dimensional prototypes $\{\mu_k(0), \sigma_k^2(0)\}_{k=1}^K$ by running K-means on $X_{\text{init}}$
3: Run Barnes–Hut t-SNE on $X_{\text{init}}$ to obtain low-dimensional points $Y(0)$
4: Compute low-dimensional centroids $\{\mu_{y,k}(0)\}$ by averaging points in $Y(0)$ per cluster
5: **for** each new sample $x_{\text{new}}$ arriving in the stream **do**
6:     Find nearest high-dimensional prototype index $k^* = \arg\min_k \|x_{\text{new}} - \mu_k\|$
7:     Initialize $y_{\text{new}}$ by placing it near $\mu_{y,k^*}$ (plus small noise)
8:     Update $\mu_{k^*}$ and $\sigma_{k^*}^2$ to incorporate $x_{\text{new}}$ in one step
9:     Compute affinities between $x_{\text{new}}$ and each $\mu_k$, normalize to get $P_k$
10:    Compute affinities between $y_{\text{new}}$ and each $\mu_{y,k}$, normalize to get $Q_k$
11:    Perform one gradient descent update on $y_{\text{new}}$ to reduce $\text{KL}(P\|Q)$
12:    Update $\mu_{y,k^*}$ to incorporate $y_{\text{new}}$ in one step

---

**Algorithm 2** Bi-RSNE

---

**Require:** High-dimensional dataset $X$, seed fraction $\alpha$, number of clusters $K$, batch size $B$, learning rate $\eta$
1: Select a random seed subset $X_{\text{init}} \subset X$ of size $\alpha|X|$
2: Compute prototypes $\{\mu_k(0), \sigma_k^2(0)\}$ via K-means on $X_{\text{init}}$
3: Run Barnes–Hut t-SNE on $X_{\text{init}}$ to get $Y(0)$ and centroids $\{\mu_{y,k}(0)\}$
4: **for** each incoming batch $\{x_i\}_{i=1}^B$ **do**
5:     Compute $B \times K$ distances between $x_i$ and prototypes $\mu_k$
6:     Assign each $x_i$ to nearest prototype index $k_i$
7:     For each cluster $k$, update $\mu_k, \sigma_k^2$ using all $x_i$ assigned to $k$
8:     Compute all high-dimensional affinities $P_{i,k}$ and low-dimensional affinities $Q_{i,k}$
9:     Perform one vectorized gradient step on all $y_i$ to minimize batch KL divergence
10:    Update low-dimensional centroids $\mu_{y,k}$ using corresponding $y_i$

---

**CIFAR100 Dataset.** The CIFAR100 dataset (Krizhevsky et al., 2009) extends CIFAR10 with 100 fine-grained classes (600 images per class), also split into 50,000 training and 10,000 testing images. We again extract 1024-dimensional features using DINOv2 ViT-L/14 and CLIP ViT-L/14 to evaluate RSNE's ability to handle increased class diversity and intra-class variability. Embedding results on the full 50,000 train set images assess how well RSNE preserves nuanced cluster structures relative to Barnes–Hut t-SNE.

**iROADS Dataset.** The iROADS dataset (Rezaei et al., 2018) comprises approximately 5,000 road-scene images collected from diverse driving environments (urban, suburban, rural). From each image, we extract a 1024-dimensional feature vector using the pre-trained DINOv2 ViT-L/14 model (Oquab et al., 2023), capturing high-level semantic information such as road type, weather conditions, and object presence. These deep features form a challenging high-dimensional input, well suited to test the scalability and representational power of our methods. We evaluate embeddings of the full set to compare t-SNE and Bi-RSNE, and we measure how well clusters corresponding to annotated road conditions are preserved.

**Climate Dataset.** The climate dataset (Muthu, 2017) contains daily measurements from 50 meteorological stations over a 10-year period (2006–2016), yielding roughly 182,500 samples. Each sample records features such as temperature, humidity, wind speed, and precipitation, resulting in a 10-dimensional input. We use monthly batches (e.g., April 2008, May 2008, June 2008) to simulate a streaming environment for Bi-RSNE, and we also evaluate full-dataset embeddings for i-RSNE. Due to the continuous nature and overlap of weather variables, this dataset presents a difficult scenario for any clustering-based embedding, making it a strict test of our recursive update mechanisms. The weather conditions are categorized into five classes: clear, partly cloudy, mostly cloudy, foggy, and rainy.

**Haxby 2001 fMRI (Visual Object Recognition).** We use the widely studied Haxby 2001 (Haxby et al., 2001) visual object recognition dataset, a block-design fMRI experiment targeting ventral temporal (VT)

cortex responses to eight categories (faces, houses, cats, bottles, scissors, shoes, chairs, scrambled pictures). The full release includes 6 subjects with 12 runs per subject; each run presents eight 24-s stimulus blocks separated by rest, acquired at TR=2.5 s (120 volumes per run). We rely on the standard Nilearn distribution of this dataset (subject 2) and extract per-volume VT-masked voxel patterns (samples × voxels). In our experiments we treat each 2.5 s volume as one sample; selecting the first 7 runs and excluding "rest" yields nearly 500 labeled samples spread over the eight categories, which is convenient for comparing t-SNE and RSNE on a modest, well-annotated neuroimaging benchmark.

## 4.2 Baseline

**Barnes–Hut t-SNE.** Barnes–Hut t-SNE (van der Maaten, 2013) accelerates the original t-SNE by using a quadtree to approximate the interactions of distant points, reducing the run time to $O(N \log N)$. Although this makes it feasible to embed large datasets in a single batch, it still requires a full re-embedding when new data points are added, preventing its use in real-time or streaming scenarios.

In contrast, our RSNE framework updates existing embeddings with only partial or batch recomputation, bridging the gap between dynamic online use cases and large-scale offline processing.

## 4.3 Evaluation Metrics

We evaluate embedding quality and computational performance using the following quantitative metrics:

**Runtime.** The total embedding time is measured as the sum of:

$$T_{\text{init}} = \text{Initialization time (e.g., Barnes–Hut t-SNE on the first 20\% of samples)}, \tag{1}$$

$$T_{\text{update}} = \text{Update time for new data points (processed individually or in batches)}, \tag{2}$$

$$T_{\text{total}} = T_{\text{init}} + T_{\text{update}}. \tag{3}$$

**Silhouette Score.** The silhouette coefficient (Rousseeuw, 1987) measures how well each point lies within its cluster versus its nearest neighboring cluster. Values range from $-1$ (poor separation) to $+1$ (well-separated clusters), with higher scores indicating clearer cluster boundaries.

**Davies–Bouldin Index.** This index (Davies & Bouldin, 1979) quantifies the average similarity between each cluster and its most similar counterpart, with lower values signifying better cluster compactness and separation.

Together, these metrics provide a holistic picture of how well the embeddings balance fidelity, cluster structure, and computational cost.

## 4.4 Implementation Details

**Hardware Configuration.** Feature extraction was performed using an NVIDIA GeForce RTX 3080 GPU with 16GB of VRAM. All other experiments, including Bi-RSNE, i-RSNE and full Barnes-Hut t-SNE evaluation, were conducted on a system equipped with an Intel Xeon CPU @2.20GHz and 52GB of RAM.

**Hyperparameters.** For both RSNE and Batch incremental RSNE (Bi-RSNE), the high-dimensional input space is set to $D = 784$, while the low-dimensional embedding space is $d = 2$. We employ K-means clustering with $K = 200$ clusters for all datasets except CIFAR100, where we use $K = 1000$ to account for its finer class granularity. The learning rate is fixed at 10 to ensure stable updates during optimization. Each update step applies a single gradient iteration to adjust the embeddings efficiently. For Bi-RSNE, a batch size of 100 is used to process multiple samples simultaneously, balancing computational efficiency with accuracy. Additionally, different initialization percentages are tested to analyze trade-offs between computational overhead and embedding quality.

### 4.5 Results

### 4.5.1 CIFAR10

We begin our evaluation on the standard CIFAR10 benchmark using features extracted by two state-of-the-art vision transformers: DINOv2 ViT-L/14 and CLIP ViT-L/14. Tables 1 and 2 report silhouette scores, Davies–Bouldin indices, and runtimes for Bi-RSNE, i-RSNE, and full Barnes–Hut t-SNE under two initialization fractions (20 % and 50 %).

**DINOv2 Features.** On DINOv2 features, Bi-RSNE with only 20 % initialization embeds all 60 000 images in 36.04 s, which is over a 6.9× speedup compared to Barnes–Hut t-SNE's 247.45 s, while achieving a silhouette of 0.5074 and a Davies–Bouldin index of 0.6894. These metrics indicate substantially tighter and more separated clusters than t-SNE's 0.4278 silhouette and 0.7659 DB. When the seed set is increased to 50 %, Bi-RSNE's runtime grows to 79.43 s but the silhouette only slightly decreases to 0.4793, with the DB remaining at 0.6962, showing that the method maintains global structure even with more extensive initialization. i-RSNE at 20 % initialization achieves the highest silhouette (0.5269) and lowest DB (0.6661) in 44.70 s, demonstrating that per-point recursive updates can marginally improve local cluster fidelity at a modest computational cost. However, at 50 % initialization i-RSNE's runtime increases to 85.03 s and its clustering metrics regress toward those of Bi-RSNE, suggesting diminishing returns for larger seed fractions in the incremental setting.

**CLIP Features.** On CLIP ViT-L/14 features (Table 2), the trends are very similar, with some notable differences in clustering tightness. Bi-RSNE at 20 % runs in 36.17 s and attains a silhouette of 0.5244 and DB of 0.6882, slightly improving over DINOv2's compactness while matching its speedup. Increasing to 50 % initialization extends runtime to 86.29 s and yields a silhouette of 0.4883 and DB of 0.7558, indicating that CLIP features form slightly looser clusters as the seed grows. i-RSNE at 20 % achieves a peak silhouette of 0.5251 in 48.24 s, closely mirroring the Bi-RSNE result and demonstrating that one-step updates suffice to preserve local structure. Full Barnes–Hut t-SNE on CLIP takes 260.92 s (silhouette 0.4493, DB 0.7903), confirming that RSNE's cluster-based initialization and local refinements generalize across different transformer embeddings, consistently delivering 6–7× speedups with minimal loss or often gains in clustering quality.

Table 1: Comparison of embedding methods on CIFAR10 features extracted with DINOv2 ViT-L/14. Metrics include silhouette score (↑), Davies–Bouldin index (↓), and runtime in seconds. Embedding visualizations are provided in Appendix A.

| Method | Init.% | Visualization | Silhouette (↑) | Davies–Bouldin (↓) | Runtime (s) |
|---|---|---|---|---|---|
| Bi-RSNE | 20% | Fig. 2a | $0.5074 \pm 0.0007$ | $0.6894 \pm 0.0001$ | **36.72** $\pm 0.32$ |
| i-RSNE | 20% | Fig. 2c | **0.5269** $\pm 0.0006$ | **0.6661** $\pm 0.0002$ | $44.70 \pm 0.22$ |
| Bi-RSNE | 50% | Fig. 2b | $0.4793 \pm 0.0003$ | $0.6962 \pm 0.0002$ | $79.43 \pm 0.81$ |
| i-RSNE | 50% | Fig. 2d | $0.4669 \pm 0.0002$ | $0.7204 \pm 0.0001$ | $85.03 \pm 0.67$ |
| Barnes-Hut t-SNE | – | Fig. 2i | $0.4278 \pm 0.0000$ | $0.7659 \pm 0.0001$ | $247.45 \pm 1.42$ |

### 4.5.2 CIFAR100

Scaling up to CIFAR100, with its 100 fine-grained classes, presents a considerably more challenging embedding task. The increased class diversity and intra-class variability demand both expressive representations and robust layout algorithms to reveal meaningful cluster structure in two dimensions.

**DINOv2 Features.** Table 3 reports detailed results on DINOv2 ViT-L/14 features for both recursive and full t-SNE methods. With only 20% initialization, Bi-RSNE embeds all 60 000 images in 158.27s, yielding a silhouette of 0.1847 and Davies–Bouldin index of 3.1302, roughly 57% of Barnes–Hut t-SNE's 278.20s runtime, but with lower cluster separation. Increasing the seed set to 50% incurs a negligible time penalty (160.42s) while boosting the silhouette to 0.2325 and tightening the DB to 2.6058, indicating a much better

Table 2: Comparison of embedding methods on CIFAR10 features extracted with CLIP ViT-L/14. Metrics include silhouette score (↑), Davies–Bouldin index (↓), and runtime in seconds. Embedding visualizations are provided in Appendix A.

| Method | Init.% | Visualization | Silhouette (↑) | Davies–Bouldin (↓) | Runtime (s) |
|---|---|---|---|---|---|
| Bi-RSNE | 20% | Fig. 2e | $0.5244 \pm 0.0006$ | **0.6882** $\pm 0.0007$ | **36.17** $\pm 0.35$ |
| i-RSNE | 20% | Fig. 2h | **0.5251** $\pm 0.0007$ | $0.7055 \pm 0.0005$ | $48.24 \pm 0.22$ |
| Bi-RSNE | 50% | Fig. 2f | $0.4883 \pm 0.0004$ | $0.7558 \pm 0.0006$ | $86.29 \pm 0.56$ |
| i-RSNE | 50% | Fig. 2g | $0.4891 \pm 0.0003$ | $0.7486 \pm 0.0008$ | $90.78 \pm 0.48$ |
| Barnes-Hut t-SNE | – | Fig. 2j | $0.4493 \pm 0.0001$ | $0.7903 \pm 0.0004$ | $260.92 \pm 1.25$ |

quality–time trade-off. At 80% initialization, Bi-RSNE completes in 167.79s and achieves a silhouette of 0.2674, surpassing full t-SNE's 0.2615, and a DB of 3.0342. In the incremental setting, i-RSNE with an 80% seed attains the highest silhouette overall (0.2751) and a DB of 2.6328 in 205.61s, demonstrating that one-step updates can further refine cluster compactness at the cost of higher runtime. Reducing the seed to 50% gives a silhouette of 0.2509 and DB of 2.4436 in 245.69s, whereas a 20% seed yields a silhouette of 0.2164 and DB of 3.9259 in 266.26s. Full Barnes–Hut t-SNE remains the slowest (278.20s) with a silhouette of 0.2615 and DB of 3.0632.

Table 3: Comparison of embedding methods on CIFAR100 features extracted with DINOv2 ViT-L/14. Metrics include silhouette score (↑), Davies–Bouldin index (↓), and runtime in seconds. Embedding visualizations are provided in Appendix A.

| Method | Init.% | Visualization | Silhouette (↑) | Davies–Bouldin (↓) | Runtime (s) |
|---|---|---|---|---|---|
| Bi-RSNE | 20% | Fig. 3a | $0.1847 \pm 0.0011$ | $3.1302 \pm 0.0008$ | **158.27** $\pm 0.39$ |
| Bi-RSNE | 50% | Fig. 3b | $0.2325 \pm 0.0008$ | $2.6058 \pm 0.0007$ | $160.42 \pm 0.75$ |
| Bi-RSNE | 80% | Fig. 3c | $0.2674 \pm 0.0003$ | $3.0342 \pm 0.0005$ | $167.79 \pm 0.93$ |
| i-RSNE | 80% | Fig. 3d | **0.2751** $\pm 0.0002$ | $2.6328 \pm 0.0005$ | $205.61 \pm 0.88$ |
| i-RSNE | 50% | Fig. 3e | $0.2509 \pm 0.0006$ | **2.4436** $\pm 0.0006$ | $245.69 \pm 0.94$ |
| i-RSNE | 20% | Fig. 3f | $0.2164 \pm 0.0010$ | $3.9259 \pm 0.0008$ | $266.26 \pm 1.17$ |
| Barnes-Hut t-SNE | – | Fig. 3m | $0.2615 \pm 0.0001$ | $3.0632 \pm 0.0005$ | $278.20 \pm 1.66$ |

**CLIP Features.** Results on CLIP ViT-L/14 features (Table 4) reveal that very small seed fractions fail to capture CLIP's semantic structure: Bi-RSNE with 20% initialization finishes in just 127.75s but yields a silhouette of 0.0001 and DB of 5.0068. A 50% seed improves silhouette to 0.0509 and DB to 4.4623 in 138.63s. With 80% initialization, Bi-RSNE matches Barnes–Hut t-SNE's silhouette (0.1324 vs. 0.1309) in 172.93s, though its DB remains higher (3.8542 vs. 3.7263). In the incremental regime, i-RSNE at 80% seed achieves a silhouette of 0.1066 and DB of 4.9388 in 197.45s, while 50% and 20% seeds yield progressively lower silhouettes (0.0647 and 0.0125) and higher DBs (4.8345 and 5.6213) in 216.13s and 259.55s, respectively. Full Barnes–Hut t-SNE on CLIP features requires 278.00s, with a silhouette of 0.1309 and DB of 3.7263. These patterns indicate that CLIP embeddings demand a larger seed set to form reliable prototypes, but that Bi-RSNE and i-RSNE can still recover comparable clustering quality at substantial runtime savings.

Interestingly, for i-RSNE, the 80% seed variant completes faster than the 20% variant, contrary to the trend observed in CIFAR10. This inversion is attributed to the significantly larger number of clusters used for CIFAR100 (K=1000). With fewer clusters (e.g., K=200 in CIFAR10), lower seed fractions usually lead to faster runtimes.

Table 4: Comparison of embedding methods on CIFAR100 features extracted with CLIP ViT-L/14. Metrics include silhouette score ($\uparrow$), Davies–Bouldin index ($\downarrow$), and runtime in seconds. Embedding visualizations are provided in Appendix A.

| Method | Init.% | Visualization | Silhouette ($\uparrow$) | Davies–Bouldin ($\downarrow$) | Runtime (s) |
|---|---|---|---|---|---|
| Bi-RSNE | 20% | Fig. 3g | $0.0001 \pm 0.0009$ | $5.0068 \pm 0.0007$ | **127.75** $\pm 0.43$ |
| Bi-RSNE | 50% | Fig. 3h | $0.0509 \pm 0.0006$ | $4.4623 \pm 0.0005$ | $138.63 \pm 0.73$ |
| Bi-RSNE | 80% | Fig. 3i | **0.1324** $\pm 0.0002$ | $3.8542 \pm 0.0008$ | $172.93 \pm 0.91$ |
| i-RSNE | 80% | Fig. 3l | $0.1066 \pm 0.0001$ | $4.9388 \pm 0.0006$ | $197.45 \pm 1.04$ |
| i-RSNE | 50% | Fig. 3k | $0.0647 \pm 0.0006$ | $4.8345 \pm 0.0007$ | $216.13 \pm 1.22$ |
| i-RSNE | 20% | Fig. 3j | $0.0125 \pm 0.0008$ | $5.6213 \pm 0.0009$ | $259.55 \pm 1.45$ |
| Barnes-Hut t-SNE | – | Fig. 3n | $0.1309 \pm 0.0001$ | **3.7263** $\pm 0.0004$ | $278.00 \pm 2.10$ |

### 4.5.3 iROADS

We evaluated RSNE on the iROADS driving-scene dataset by embedding the full set of 4 656 images using 1024-dimensional DINOv2 ViT-L/14 features. Table 5 reports results for both Bi-RSNE and i-RSNE at 20% and 50% initialization alongside full Barnes–Hut t-SNE. With no seed, Barnes–Hut t-SNE requires 9.96 s to produce an embedding with silhouette 0.3038 and Davies–Bouldin (DB) index 1.0085. These moderate clustering metrics reflect the dataset's seven annotated scene conditions, but come at the cost of nearly 10 s of compute time.

Bi-RSNE with a 20 % seed set cuts runtime to just 3.26 s, over three times faster, while dramatically improving cluster separation (silhouette 0.6852, DB 0.4369). In comparison, i-RSNE with the same 20 % seed runs in 4.13 s (still more than twice as fast as full t-SNE) and yields a silhouette of 0.6302 and DB 0.5041, demonstrating that one-step, point-wise updates can also produce tight clusters with only a modest additional cost. Increasing the seed fraction to 50 % raises Bi-RSNE's runtime to 6.02 s (silhouette 0.4281, DB 0.8094), still a 40 % speedup over t-SNE. In that setting, i-RSNE completes in 7.31 s with silhouette 0.4113 and DB 0.8344, indicating that larger seeds diminish the benefit of per-point updates but remain faster than a full re-embedding.

We also test a setting where $N - 1$ points were used for initialization, and only one incremental update was performed for the remaining sample. This provides a stringent test of whether the initialization and update mechanism can approximate, or even surpass, the quality of embeddings obtained by full t-SNE optimization. Quantitative results on DINOv2 features of the iROADS dataset reveal that the incremental approach not only matches but slightly improves the embedding quality. Specifically, our method achieved a Silhouette score of 0.3092 compared to 0.3088 from full t-SNE, and a Davies–Bouldin Index of 1.0007 compared to 1.0070. These results indicate that even with a single update step, the incremental procedure preserves inter-cluster separation while improving intra-cluster compactness. This demonstrates that the proposed initialization and update strategy can provide embeddings that are at least as reliable as those from the standard full t-SNE run, while being more efficient and better aligned with continual or streaming data scenarios.

Overall, these iROADS results demonstrate that both RSNE variants exploit clear semantic modalities in a modestly sized, real-world dataset: Bi-RSNE's batch updates achieve the best clustering metrics at minimal runtime, while i-RSNE's one-step refinements offer a middle ground, which is more accurate than Barnes–Hut t-SNE and easily adaptable to streaming data, with only a slight time penalty relative to Bi-RSNE.

### 4.5.4 fMRI (Haxby 2001)

We evaluated RSNE on the Haxby visual object recognition fMRI dataset using the standard ventral–temporal (VT) mask and per-volume features across the eight object categories. Under the same preprocessing and initialization protocol as our t-SNE baseline, both Bi-RSNE and i-RSNE achieved clear runtime reductions while preserving overall embedding quality: silhouette and Davies–Bouldin scores remained similar to

Table 5: Comparison of embedding methods on iROADS (4 656 samples) with DINOv2 ViT-L/14 features. Metrics include silhouette score (↑), Davies–Bouldin index (↓), and runtime in seconds. Visualizations are in Appendix A.

| Method | Init.% | Visualization | Silhouette (↑) | Davies–Bouldin (↓) | Runtime (s) |
|---|---|---|---|---|---|
| Bi-RSNE | 20% | Fig. 4a | **0.6852** ±0.0003 | **0.4369** ±0.0005 | **3.26** ±0.12 |
| i-RSNE | 20% | Fig. 4c | $0.6302 \pm 0.0003$ | $0.5041 \pm 0.0004$ | $4.13 \pm 0.08$ |
| Bi-RSNE | 50% | Fig. 4b | $0.4281 \pm 0.0002$ | $0.8094 \pm 0.0005$ | $6.02 \pm 0.35$ |
| i-RSNE | 50% | Fig. 4d | $0.4113 \pm 0.0001$ | $0.8344 \pm 0.0006$ | $7.31 \pm 0.42$ |
| Barnes–Hut t-SNE | – | Fig. 4e | $0.3038 \pm 0.0001$ | $1.0085 \pm 0.0002$ | $9.96 \pm 0.77$ |

Barnes–Hut t-SNE. This outcome is consistent that VT representations are distributed and overlapping, which typically limits strong cluster separation; nevertheless, RSNE maintains the same qualitative structure at lower computational cost, making it practical for rapid fMRI exploration and iterative analyses.

Table 6: Comparison of embedding methods on Haxby fMRI (subject 2, VT mask; ∼7 runs, non-rest volumes).

| Method | Init.% | Silhouette (↑) | Davies–Bouldin (↓) | Runtime (s) |
|---|---|---|---|---|
| Bi-RSNE | 20% | $-0.0977 \pm 0.0096$ | $26.1182 \pm 2.3087$ | **0.486** ±0.052 |
| i-RSNE | 20% | $-0.0851 \pm 0.0061$ | $25.7151 \pm 2.1954$ | $0.642 \pm 0.066$ |
| Bi-RSNE | 50% | **-0.0765** ±0.0056 | $25.6565 \pm 1.7633$ | $0.880 \pm 0.038$ |
| i-RSNE | 50% | $-0.0779 \pm 0.0047$ | **26.1999** ±1.6205 | $1.065 \pm 0.031$ |
| Barnes–Hut t-SNE | – | $-0.0831 \pm 0.0042$ | $25.4699 \pm 1.2609$ | $1.202 \pm 0.102$ |

### 4.5.5 Streaming Data Integration on Climate Dataset

To evaluate RSNE's performance in an online setting, we simulated a streaming scenario on the Climate dataset, which comprises approximately 182 500 daily measurements from 50 meteorological stations over ten years. We first embedded 20 % of the data with Barnes–Hut t-SNE to establish an initial layout, then applied Bi-RSNE and i-RSNE to incorporate three subsequent monthly batches (April–June 2008). Table 7 and Figure 1 summarize the quantitative and qualitative results.

Full re-embedding of all 182 500 samples via Barnes–Hut t-SNE requires 508.09 s, yielding a silhouette of –0.0676 and a Davies–Bouldin index of 8.2827. These metrics reflect the inherently overlapping nature of climate variables, where distinct clusters are difficult to separate. By contrast, Bi-RSNE with only a 20 % seed reduces total runtime to 83.79 s (an 83 % reduction), while producing a silhouette of –0.0806 and a Davies–Bouldin index of 8.9379. Although the silhouette degrades slightly and DB increases modestly, the quality–time trade-off is highly favorable for interactive or iterative analyses. In the same 20 % setting, i-RSNE embeds the stream in 147.14 s with silhouette –0.0803 and DB 8.9509, showing that one-step, point-wise updates can nearly match Bi-RSNE's clustering metrics at the cost of additional runtime.

When the seed fraction is increased to 50 %, Bi-RSNE's runtime grows to 228.49 s with silhouette –0.0911 and DB 5.7680, indicating that a larger seed can capture slightly more compact clusters (lower DB) at the expense of higher compute. In this setting, i-RSNE completes in 268.40 s and achieves silhouette –0.0916 and DB 5.7662, closely matching Bi-RSNE's DB but at a roughly 18 % longer runtime. Even with a 50 % seed, both RSNE variants remain nearly 45 % faster than full t-SNE, demonstrating RSNE's flexibility in balancing speed against embedding fidelity.

Figure 1 visualizes the embedding progression: the top panel shows the initial 20 % layout, and the subsequent panels illustrate the integration of each monthly batch. New points (outlined in black) slot into their nearest cluster prototypes with minimal distortion of the existing structure, highlighting the stability and responsiveness of both Bi-RSNE and i-RSNE update mechanisms.

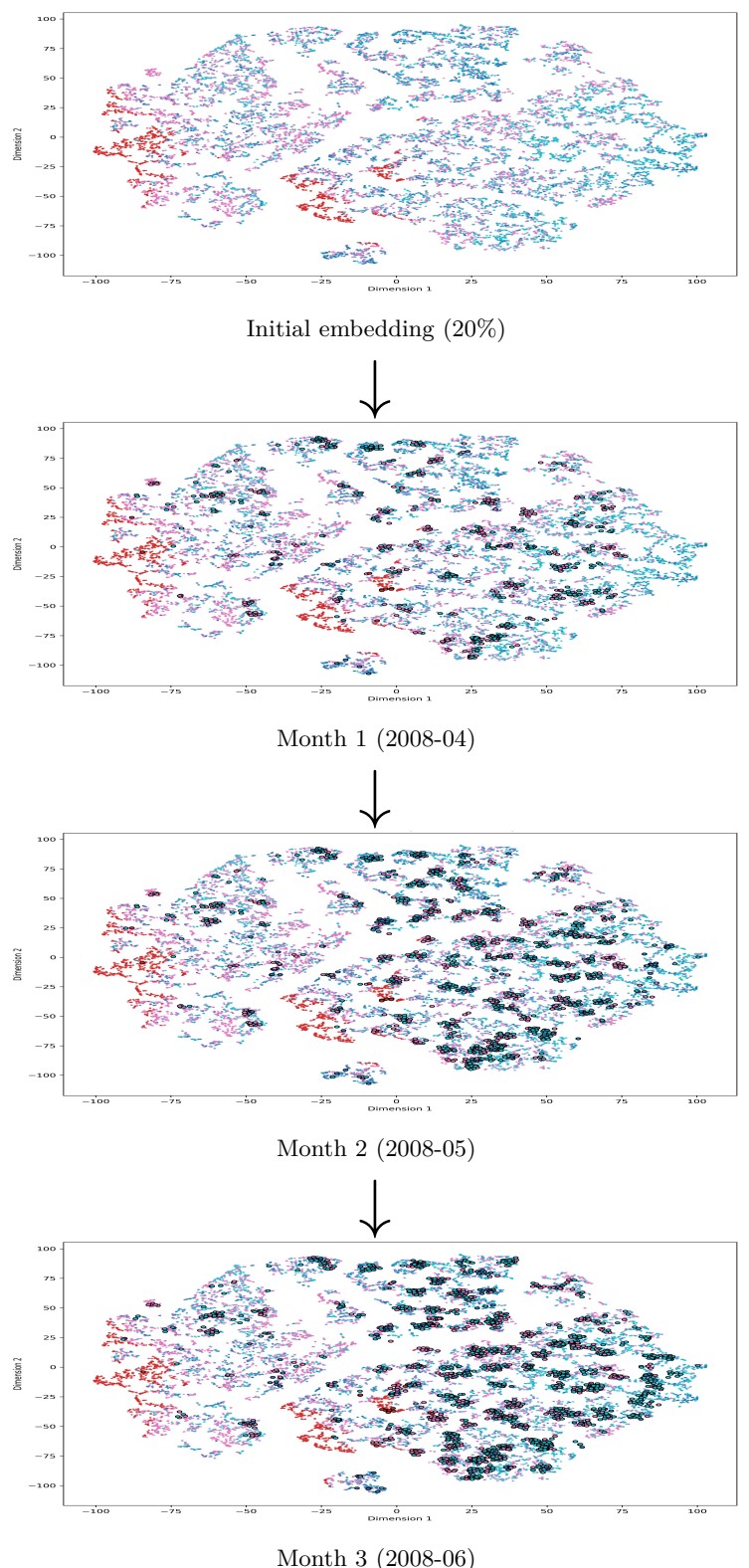

Initial embedding (20%)

Month 1 (2008-04)

Month 2 (2008-05)

Month 3 (2008-06)

Figure 1: Progression of Bi-RSNE embeddings over time on the Climate dataset. Starting from an initial 20 % layout, three monthly batches are integrated incrementally. New points (outlined) are assigned into existing clusters with minimal perturbation.

Table 7: Streaming integration performance on the Climate dataset. Metrics include silhouette score ($\uparrow$), Davies–Bouldin index ($\downarrow$), and total runtime.

| Method | Init.% | Silhouette ($\uparrow$) | Davies–Bouldin ($\downarrow$) | Runtime (s) |
|--------|--------|------------------------|-------------------------------|-------------|
| Bi-RSNE | 20% | $-0.0806 \pm 0.0005$ | $8.9379 \pm 0.0012$ | **83.79** $\pm 0.51$ |
| i-RSNE | 20% | $-0.0803 \pm 0.0004$ | $8.9509 \pm 0.00013$ | $147.14 \pm 0.62$ |
| Bi-RSNE | 50% | $\check{}0.0911 \pm 0.0002$ | $5.7680 \pm 0.0009$ | $228.49 \pm 1.02$ |
| i-RSNE | 50% | $-0.0916 \pm 0.0002$ | **5.7662** $\pm 0.0010$ | $268.40 \pm 1.31$ |
| Barnes–Hut t-SNE | – | **$-0.0676$** $\pm 0.0001$ | $8.2827 \pm 0.0008$ | $508.09 \pm 3.45$ |

## 4.6 Discussion

### 4.6.1 Embedding Quality in Foundational Model Feature Spaces

RSNE is well suited for visualizing high-dimensional features from foundational models like DINOv2 and CLIP. These models produce embeddings that are often organized in meaningful groupings, such as object classes in DINOv2 or semantic similarities in CLIP. RSNE captures these groupings effectively by first placing representative samples using clustering and then adjusting the rest of the points in a way that respects the overall structure.

One of the main advantages of RSNE in this context is that it keeps the global layout stable while gradually adding new points. This makes it easier to see consistent patterns across different types of features, even when those features vary in how tightly they group (e.g., more compact in DINOv2, more spread out in CLIP). RSNE helps maintain this structure with much less computation than re-running full t-SNE, making it especially useful when working with large embedding sets or during model development, where updates and comparisons are frequent.

## 4.7 Ablations

Table 8: Ablation over gradient steps and learning rate $\eta$, performed on the iROADS dataset with DINOv2 ViT-L/14 features and 20% initialization size. Higher Silhouette is better; lower DB Index is better. Initialization time was between 2.0–2.8s depending on $\eta$. Reported runtimes are incremental update times only.

| $\eta$ | Steps | Time (s) | Silhouette | DB Index |
|--------|-------|----------|------------|----------|
| 10 | 1 | 0.7 | 0.6530 | 0.4966 |
| | 5 | 0.7 | 0.6378 | 0.5120 |
| | 10 | 0.9 | 0.6375 | 0.5123 |
| | full | 4.6 | 0.6442 | 0.5056 |
| 1 | 1 | 0.8 | 0.6858 | 0.4207 |
| | 5 | 0.9 | 0.6937 | 0.4097 |
| | 10 | 1.0 | 0.6963 | 0.4052 |
| | full | 4.1 | 0.6912 | 0.4079 |
| 0.1 | 1 | 0.8 | 0.6757 | 0.4639 |
| | 5 | 0.8 | 0.6763 | 0.4630 |
| | 10 | 1.0 | 0.6776 | 0.4610 |
| | full | 5.5 | 0.6884 | 0.4407 |
| 0.01 | 1 | 0.5 | 0.6614 | 0.4689 |
| | 5 | 0.8 | 0.6607 | 0.4698 |
| | 10 | 0.9 | 0.6604 | 0.4700 |
| | full | 5.5 | 0.6654 | 0.4635 |

We conducted an ablation study to examine the effect of two key hyperparameters: the number of gradient descent steps per incremental update and the learning rate $\eta$. Results on the iROADS dataset with DINOv2 ViT-L/14 features and 20% initialization size are summarized in Table 8. Across all settings, we observe that the method is relatively stable with respect to the number of gradient steps. In fact, even a single step yields embeddings of comparable quality to full convergence, confirming that the proposed prototype-based initialization provides a strong starting point that requires only minor refinement.

The learning rate $\eta$ also influences performance. While very small values (e.g., $\eta = 0.01$) slow convergence and provide limited gains, moderate values ($\eta = 1$) achieve the best overall trade-off, delivering the highest Silhouette scores and lowest DB indices with efficient runtime. Larger values ($\eta = 10$) are more sensitive, sometimes degrading quality slightly despite reduced time per update. These results validate the robustness of our approach and highlight that a modest learning rate with minimal update steps is sufficient to achieve strong embedding quality with low computational cost.

We further assessed how prototypes are chosen for Bi-RSNE on iROADS, comparing centroids from $k$-means ("clustering") to uniformly sampled data points ("random"). Using the same 20% and 50% initialization sizes, clustering-based prototypes consistently yielded better separation (higher Silhouette, lower DB) with only a small runtime overhead relative to random selection (Table 9). Intuitively, clustering produces centroids that minimize within-cluster squared distortion and act as effective anchors that cover dense regions of the feature space, yielding more stable nearest-prototype assignments and better preservation of local neighborhoods in the prototype-affinity step of RSNE; random prototypes lack this coverage and increase assignment noise.

Table 9: Prototype selection ablation on iROADS (DINOv2 ViT-L/14): Bi-RSNE with clustering vs. random prototypes under two initialization sizes. Metrics are mean ± std.

| Prototype selection | Init.% | Silhouette (↑) | Davies–Bouldin (↓) | Runtime (s) |
|---|---|---|---|---|
| Random | 20% | $0.6586 \pm 0.0004$ | $0.4662 \pm 0.0006$ | $3.12 \pm 0.09$ |
| Clustering | 20% | $\mathbf{0.6852} \pm 0.0003$ | $\mathbf{0.4369} \pm 0.0005$ | $3.26 \pm 0.12$ |
| Random | 50% | $0.4132 \pm 0.0002$ | $0.8203 \pm 0.0004$ | $5.88 \pm 0.12$ |
| Clustering | 50% | $\mathbf{0.4281} \pm 0.0002$ | $\mathbf{0.8094} \pm 0.0005$ | $6.02 \pm 0.35$ |

## 4.8 Overall Findings and Implications

Our results highlight several key advantages of the RSNE framework across a range of datasets and embedding types.

- **Faster, High-Quality Mapping:** Bi-RSNE provides a clear trade-off between runtime and embedding quality. By adjusting the initialization fraction, users can significantly reduce computation time, often by 50% or more, while still achieving results that are close in quality to full t-SNE. This makes RSNE well suited for fast, large-scale visualization.

- **Efficient Streaming Integration:** i-RSNE supports incremental updates by embedding new points individually without reprocessing the entire dataset. This is useful for applications where new data arrives continuously, such as monitoring or interactive data exploration.

- **Consistency Across Embeddings:** RSNE works reliably across different feature types. Whether using DINOv2 or CLIP embeddings, it captures the main structure of the data while preserving both local and global relationships. This makes it a flexible tool for analyzing features from various deep learning models.

- **Scalability and Practical Use:** Because RSNE updates depend only on a small portion of the data and a limited number of steps, it remains fast and efficient even as datasets grow. It avoids the high computational cost of traditional t-SNE, making it practical for both offline and real-time settings.

## 5  Limitations

Although RSNE achieves clear runtime advantages over Barnes–Hut t-SNE, a few limitations should be noted. First, the efficiency of i-RSNE can degrade when the number of prototypes $K$ is very large, since each new sample must sequentially compute affinities against all prototypes. This behavior is visible in Table 3 ($K = 1000$), where i-RSNE becomes slower relative to Bi-RSNE. In contrast, Bi-RSNE processes entire batches in a vectorized manner, amortizing prototype comparisons across many points and maintaining faster runtimes. For smaller prototype sizes (e.g., $K = 100$ in Tables 1–2), this effect is much less pronounced.

Second, like other prototype-based methods, RSNE assumes some degree of cluster structure in the data. When the structure is weak, prototype initialization is less effective, though our ablation with random prototypes (Table 9) shows that the method still degrades gracefully and remains efficient.

## 6  Acknowledgement

This work is supported by ELSA– European Lighthouse on Secure and Safe AI funded by the European Union under grant agreement No. 101070617. Views and opinions expressed are, however, those of the author(s) only and do not necessarily reflect those of the European Union or European Commission. Neither the European Union nor the European Commission can be held responsible.

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

# A  Appendix

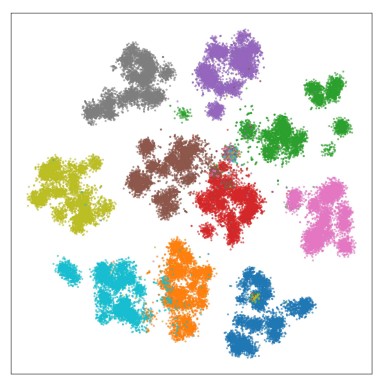

(a) Bi-RSNE (20%) on DINOv2 ViT-L/14 features

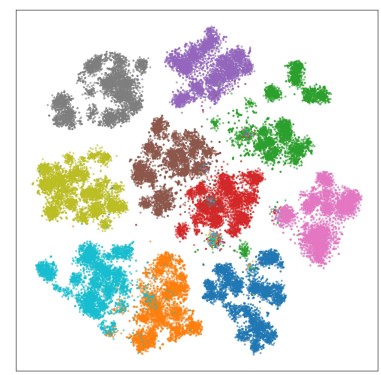

(b) Bi-RSNE (50%) on DINOv2 ViT-L/14 features

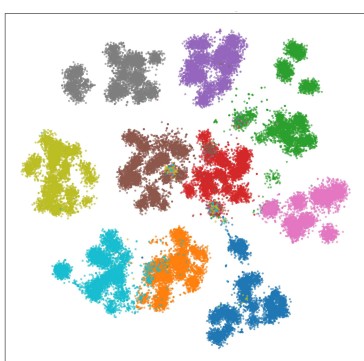

(c) i-RSNE (20%) on DINOv2 ViT-L/14 features

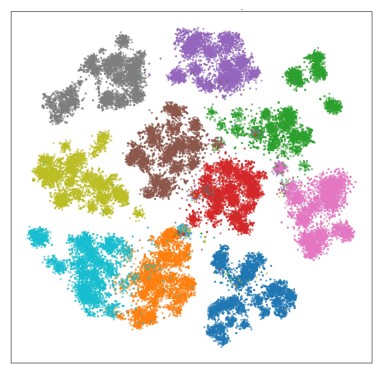

(d) i-RSNE (50%) on DINOv2 ViT-L/14 features

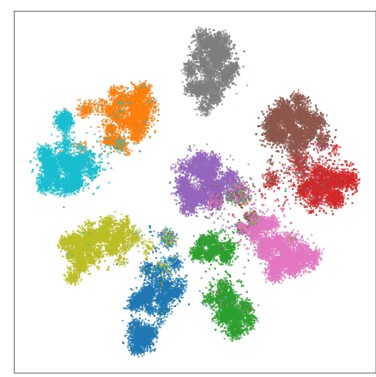

(e) Bi-RSNE (20%) on CLIP ViT-L/14 features

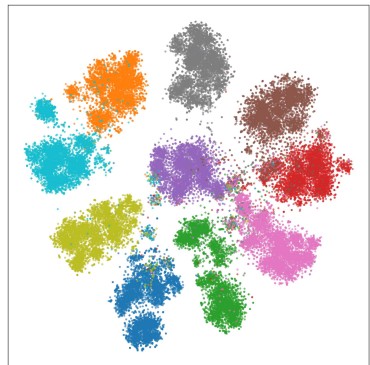

(f) Bi-RSNE (50%) on CLIP ViT-L/14 features

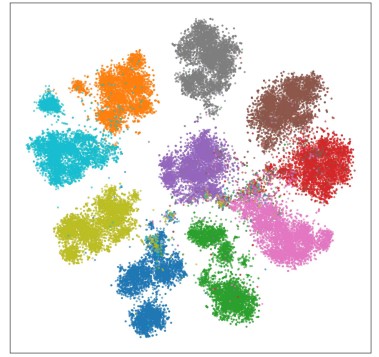

(g) i-RSNE (50%) on CLIP ViT-L/14 features

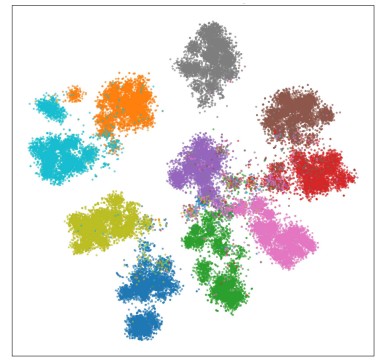

(h) Bi-RSNE (20%) on CLIP ViT-L/14 features

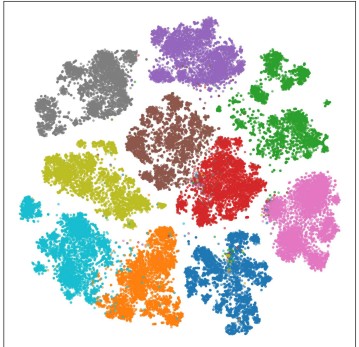

(i) Barnes-Hut t-SNE on DINOv2 ViT-L/14 features

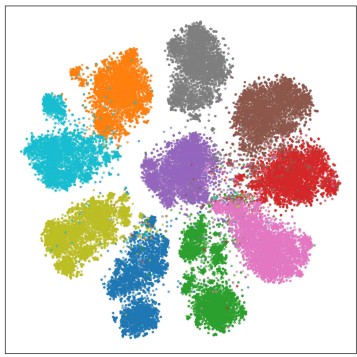

(j) Barnes-Hut t-SNE on CLIP ViT-L/14 features

Figure 2: 2D embeddings of CIFAR10 features extracted with DINOv2 and CLIP ViT-L/14 under varying initialization ratios using Bi-RSNE, i-RSNE and full Barnes-Hut t-SNE.

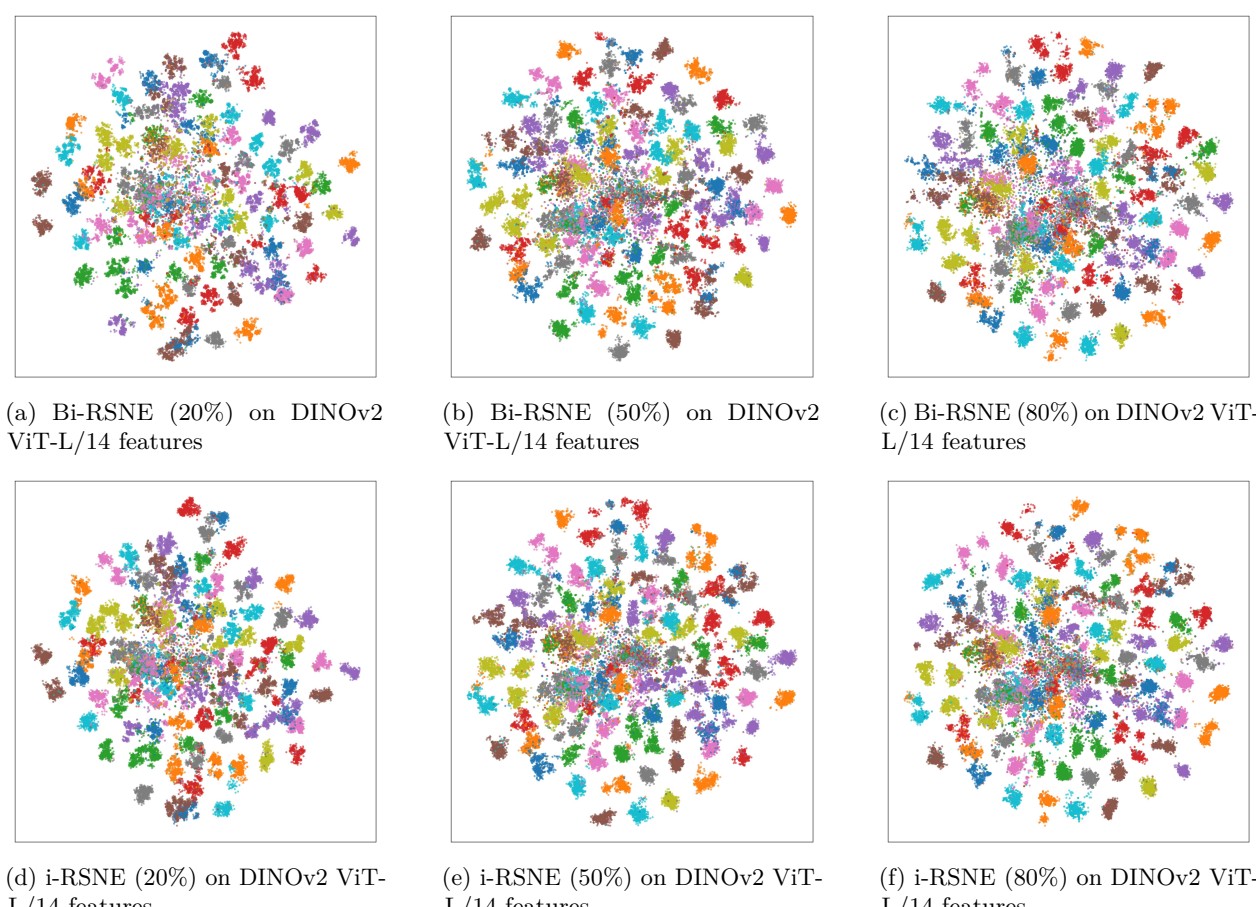

(a) Bi-RSNE (20%) on DINOv2 ViT-L/14 features

(b) Bi-RSNE (50%) on DINOv2 ViT-L/14 features

(c) Bi-RSNE (80%) on DINOv2 ViT-L/14 features

(d) i-RSNE (20%) on DINOv2 ViT-L/14 features

(e) i-RSNE (50%) on DINOv2 ViT-L/14 features

(f) i-RSNE (80%) on DINOv2 ViT-L/14 features

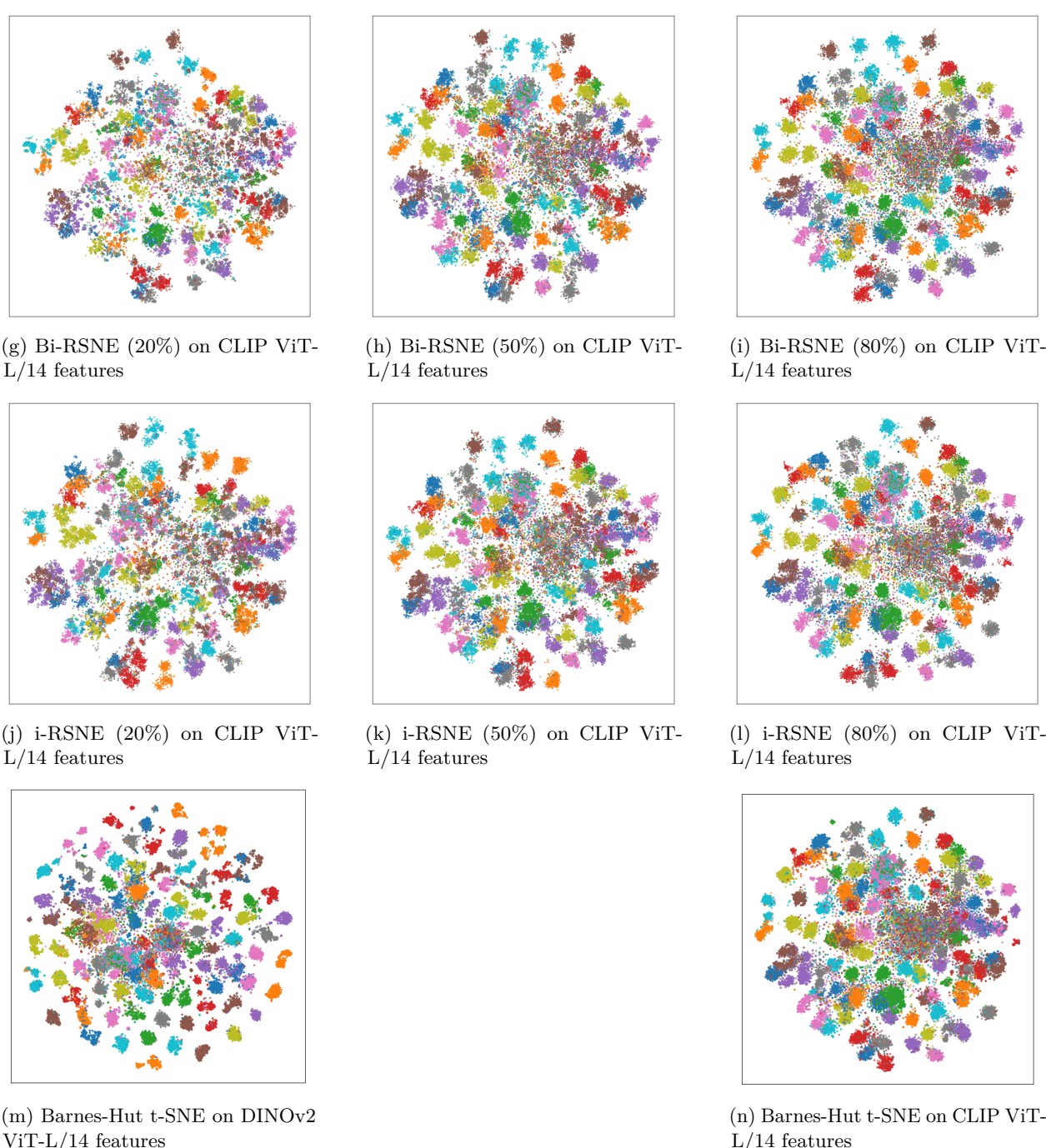

(g) Bi-RSNE (20%) on CLIP ViT-L/14 features

(h) Bi-RSNE (50%) on CLIP ViT-L/14 features

(i) Bi-RSNE (80%) on CLIP ViT-L/14 features

(j) i-RSNE (20%) on CLIP ViT-L/14 features

(k) i-RSNE (50%) on CLIP ViT-L/14 features

(l) i-RSNE (80%) on CLIP ViT-L/14 features

(m) Barnes-Hut t-SNE on DINOv2 ViT-L/14 features

(n) Barnes-Hut t-SNE on CLIP ViT-L/14 features

Figure 3: 2D embeddings of CIFAR100 features extracted with DINOv2 and CLIP ViT-L/14 under varying initialization ratios using Bi-RSNE, i-RSNE and full Barnes-Hut t-SNE.

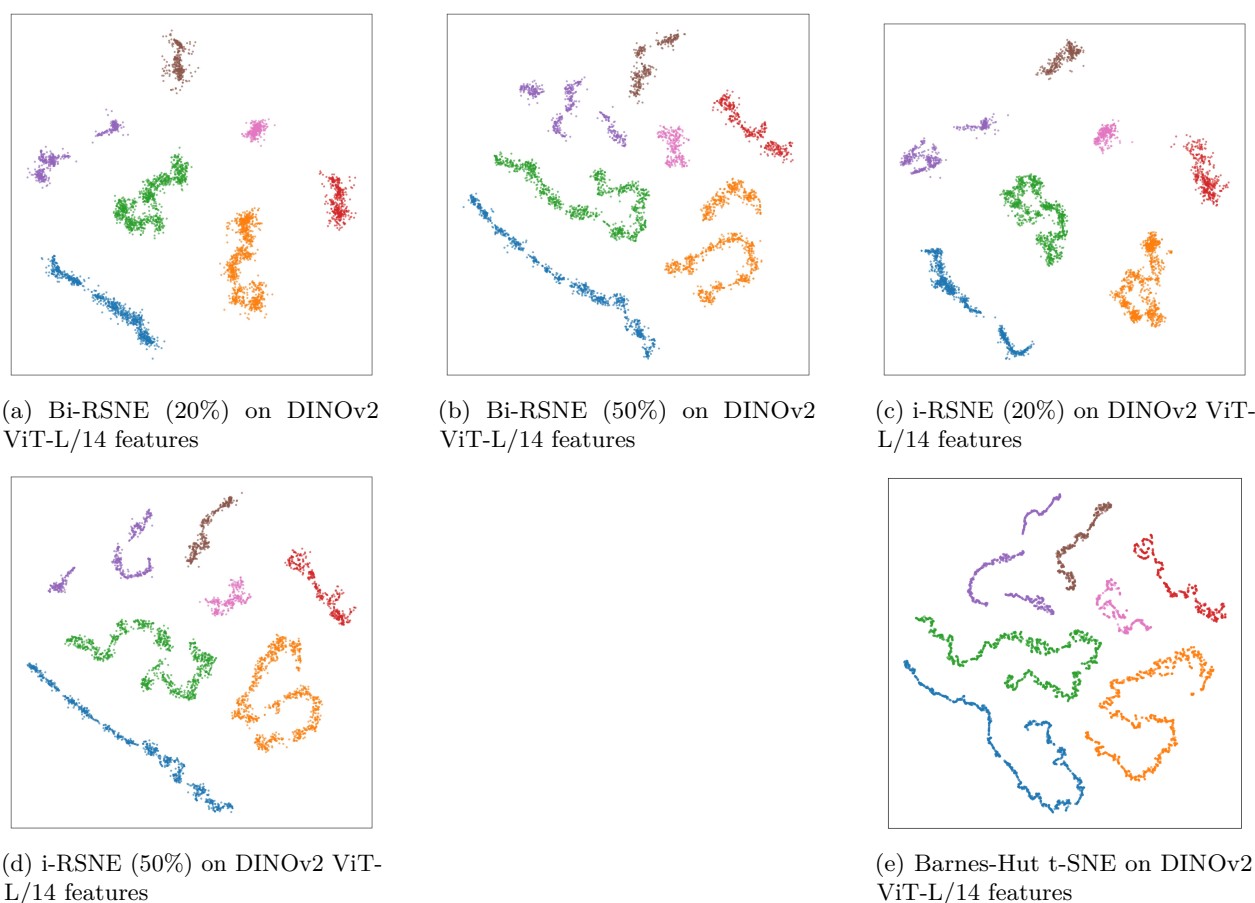

(a) Bi-RSNE (20%) on DINOv2 ViT-L/14 features

(b) Bi-RSNE (50%) on DINOv2 ViT-L/14 features

(c) i-RSNE (20%) on DINOv2 ViT-L/14 features

(d) i-RSNE (50%) on DINOv2 ViT-L/14 features

(e) Barnes-Hut t-SNE on DINOv2 ViT-L/14 features

Figure 4: 2D embeddings of iROADS features extracted with DINOv2 ViT-L/14 under varying initialization ratios using Bi-RSNE, i-RSNE and full Barnes-Hut t-SNE.

