# OpenReview forum: "Recursive SNE: Fast Prototype-Based t-SNE for Large-Scale and Online Data"
_TMLR — Accepted by TMLR_

### Review · Reviewer_KBDs · 2025-07-02

**Summary Of Contributions:**

In this paper, the authors propose an economic version of t-SNE, which also allows to handle streaming data. The main idea is to select a subset of the original dataset, and run t-SNE on this data, while in parallel computing k-means clusters in the original space. When a new point is to be embedded, the closest centroid in the high-dimensional space is computed, and the low-dimensional vector of the new point is initialized in the corresponding low-dimensional centroid. Then one step of gradient descent of the Kullback-Leibler divergence (of the preference of the new point to clusters in the high and low dimensional spaces).

A version for batch processing is also presented.

The methods are tested in CIFAR10/100, iROADS, and a Climate Dataset. The 2d visualizations are shown, and the Silhouette score and Davies-Boulding Index are reported. The results are very good in general.

**Audience:**

Yes

**Claims And Evidence:**

Yes

**Requested Changes:**

## Critical

 - I was expecting a couple of ablation studies, related to each other. Namely:
	 - You do only one GD step. How does the number of steps affect the result?
	 - Maybe more important, and related to one point mentioned above. Would it be desirable to minimize the KL all the way? (GD until convergence) I would like to see some results and comparisons related to this. If the results are worse, what is the intuition behind he proposed method working

 - What happens when the original data is not clustered? For instance, consider the example mentioned in the paper: streaming resting-state fMRI data for a patient.

 - Table 3 shows that for Bi-RSNE, increasing the size of the seed set impacts the runtime, suggesting that the clustering step is the most time-consuming part.
On the other hand, for i-RSNE the behaviour is reversed.
A follow-up experiment on this would be nice.

 - Also, based on this last fact, I wonder whether for larger datasets the runtime of (B)i-RSNE is still better than Barnes-Hut t-SNE.

 - On page 7, first paragraph. The runtime of Bi-RSNE is O(BK(D+d)). Please detail this order. Also, what is the (total) order for i-RSNE? It only says O(Kd) per update, and with these orders, I don't see why doing the batch procedure is way better than doing B times the i-RSNE. Is it the order? The constant?

## Strengthen the work

 - Since, at the end of the day, this is a method to use as an alternative to pure t-SNE, it would also be nice to include other methods in the comparison (UMAP, and other methods mentioned in the paper).

 - Add a paragraph for setting the notation throughout the paper. For instance, $x$ will be used for the original datapoint in the original dimension $D$, and $y$ for the embeddings, etc. The notation $d$ for the dimension of the embedding space is defined at the end of the presentation of i-RSNE!

 - On page 6, it says "while leaving the rest of the embeddings essentially unchanged". Why "essentially"? The GD update only affects the embedding of the new point, right?

  - A reorganization of Section 2 may be needed. Also, based on Section 2.3, I was expecting an application to fMRI data.

**Strengths And Weaknesses:**

Strengths
The paper is generally well-written, the idea is simple, and the results are promising.


Weaknesses

The idea of running only one GD step is not justified. I understand it in the context of efficiency. However, what is the intuition behind it? Is this an approximation of another method?

---

> ### Author Response · Authors · 2025-08-19
>
> We thank the Reviewer for the positive comments and suggestions.
>
> >Maybe more important, and related to one point mentioned above. Would it be desirable to minimize the KL all the way? (GD until convergence) I would like to see some results and comparisons related to this. If the results are worse, what is the intuition behind he proposed method working
>
> We have included an ablation table analyzing the effect of different GD steps. Our results show that a single step already achieves stable performance due to the initialization near the centroid. Additional steps lead to minimal improvement but substantially increase runtime, which justifies our design choice.
>
> >What happens when the original data is not clustered? For instance, consider the example mentioned in the paper: streaming resting-state fMRI data for a patient.
>
> Since RSNE relies on a prototype selection step based on clustering, it inherently assumes that the data exhibits some cluster structure. If the data were not clustered, prototype selection would lose its meaning, and the initialization for new points would be less effective. In our experiments (iROADS, CIFAR10/100, Climate), the datasets exhibited sufficient inherent structure to enable effective clustering-based prototype selection.
>
> In particular, even though the Climate dataset was notably weakly separable, as reflected in its low Silhouette and high Davies-Bouldin scores with full t-SNE (Table 6), we still observed that RSNE maintained comparable embedding quality while running much faster.
>
> >Table 3 shows that for Bi-RSNE, increasing the size of the seed set impacts the runtime, suggesting that the clustering step is the most time-consuming part. On the other hand, for i-RSNE the behaviour is reversed. A follow-up experiment on this would be nice.
>
> >On page 7, first paragraph. The runtime of Bi-RSNE is O(BK(D+d)). Please detail this order. Also, what is the (total) order for i-RSNE? It only says O(Kd) per update, and with these orders, I don't see why doing the batch procedure is way better than doing B times the i-RSNE. Is it the order? The constant?
>
> Although both methods share prototype-related costs, **Bi-RSNE is more efficient for batch updates**:
>
> $$
> T_{\mathrm{Bi}}(B) = O\bigl(BK(D + d)\bigr),
> $$
>
> since it processes all new points simultaneously and benefits from vectorized computations.
>
> In contrast, sequential **i-RSNE** processes each of the \(B\) points individually:
>
> $$
> T_{\mathrm{i}}(B) = B \cdot O(Kd),
> $$
>
> incurring computation per point. Vectorized batch updates inherently enable parallel computation across CPU threads. Instead of processing each sample sequentially, operations like distance calculations or gradient updates are applied to the entire batch in a single call.
>
> For the CIFAR100 experiments using large number of prototypes 𝐾=1000 as shown in Table 3, Bi-RSNE consistently outperforms i-RSNE in runtime across seed sizes because it performs its most expensive computations, which are distance and affinity calculations against all prototypes, all at once in parallel batch operations. In contrast, i-RSNE handles each new data point sequentially, performing the same expensive large number of prototype computations without benefiting from parallel execution.
>
> >Add a paragraph for setting the notation throughout the paper. For instance, x will be used for the original datapoint in the original dimension D, and y for the embeddings, etc. The notation d for the dimension of the embedding space is defined at the end of the presentation of i-RSNE!
>
> We revised the paper to set clear notation upfront:  $x$ for original data in $D$-dimensional space and $y$ for embeddings in $d$-dimensional space.
>
> >On page 6, it says "while leaving the rest of the embeddings essentially unchanged". Why "essentially"? The GD update only affects the embedding of the new point, right?
>
> Thanks for the observation! Yes, GD update only affects new points' embeddings. We rephrased this to state explicitly: “The GD update only affects the embedding of the new point, while the embeddings of previous points remain fixed".
>
> >Also, based on Section 2.3, I was expecting an application to fMRI data.
>
> We were unable to apply RSNE to fMRI datasets due to the large data volumes and the resulting computational and time constraints exceeding our available resources. Nevertheless, we appreciate this suggestion and agree that applying RSNE to high-dimensional neuroimaging data would be a valuable.

---

> > ### Comment · Reviewer_KBDs · 2025-08-28
> >
> > Thank you for the response and for including the ablation study on GD convergence, as well as the notation paragraph.
> > There are still a few points that remain either unanswered or not fully addressed in the revised version.
> >
> > Regarding my question, “What happens when the original data is not clustered?”:
> > I understand that “prototype selection would lose its meaning” (which is why I asked in the first place), but will the method still work? Will it perform worse—and if so, by how much? There is no additional experiment analyzing this. Is there any correlation between performance and a measure of the group structure in the data?
> >
> > Similarly, there is no follow-up experiment on the impact of the seed set size. From the response, I still don’t know what to expect regarding the performance of the two methods in such scenarios.
> >
> > I also could not find the discussion of the orders in the revised version, nor their explanation. For example, why does one version depend on the dimension D while the other does not?
> >
> > Finally, I have seen several papers apply dimensionality reduction techniques to fMRI data, and I have even run some myself on a personal computer. If this paper is a faster version of t-SNE, why would this be a problem? If it is indeed a limitation, then the reference to fMRI data in Section 2 should be removed, and the paper should clearly state its limitations.

---

> > > ### Author Response · Authors · 2025-09-10
> > >
> > > >Regarding my question, “What happens when the original data is not clustered?”: I understand that “prototype selection would lose its meaning” (which is why I asked in the first place), but will the method still work? Will it perform worse—and if so, by how much? There is no additional experiment analyzing this. Is there any correlation between performance and a measure of the group structure in the data?
> > >
> > > We added Table 9 in the revised paper, where the only practical way we could think of simulating such a scenario was to use random samples as prototypes instead of clustering, keeping the number of random samples equal to K (the number of clusters) for a fair comparison. We applied Bi-RSNE on iROADS dataset; Bi-RSNE still runs efficiently, but the separation quality is weaker. By contrast, clustering-based prototypes give clearer separation with only a very small runtime overhead, showing that RSNE works in both cases but benefits noticeably when the data have some underlying structure.
> > >
> > > >Similarly, there is no follow-up experiment on the impact of the seed set size. From the response, I still don’t know what to expect regarding the performance of the two methods in such scenarios.
> > >
> > > We have explicitly added the sensitivity of i-RSNE to large seed sizes as a limitation in the newly included Limitations section. While we did not add a new experiment, we clarified in the revised paper that the effect of seed size is already visible in Table 3, where K=1000. In this setting, i-RSNE processes each new point sequentially and repeats the large prototype computations K times per sample, whereas Bi-RSNE bypasses this cost by exploiting vectorized batch operations. This explains why runtime diverges between the two methods as the seed size grows in Table 3, but not in Tables 1–2 (where K=100 and the effect is negligible).
> > >
> > > >I also could not find the discussion of the orders in the revised version, nor their explanation. For example, why does one version depend on the dimension D while the other does not?
> > >
> > > We thank the Reviewer for this observation. In the revised paper we have added the full order expressions for both i-RSNE and Bi-RSNE at the end of the methodology section. In the earlier version we mistakenly described i-RSNE as depending only on d, but it also involves D; with this correction, both methods share the same asymptotic complexity, though Bi-RSNE achieves faster runtime in practice by processing batches in parallel, whereas i-RSNE updates points sequentially.
> > >
> > > >Finally, I have seen several papers apply dimensionality reduction techniques to fMRI data, and I have even run some myself on a personal computer. If this paper is a faster version of t-SNE, why would this be a problem? If it is indeed a limitation, then the reference to fMRI data in Section 2 should be removed, and the paper should clearly state its limitations.
> > >
> > > For applying on fMRI, we used RSNE on the Haxby 2001 visual-object dataset (8 categories), sampling nearly 500 non-rest volumes from the first 7 runs; on this benchmark RSNE achieved similar quality as t-SNE with reduced runtime, addressing feasibility on typical neuroimaging data. We added all necessary information about preprocessing on this dataset and results in the revised paper.

---

### Review · Reviewer_Mg1i · 2025-07-09

**Summary Of Contributions:**

The paper presents a variant of the t-SNE algorithm for dimensionality reduction that is designed for fast incremental updates with applications for large-scale datasets and streaming data.  The authors state their new algorithm and evaluate its performance in terms of both computational efficiency and clustering performance on various datasets.  The results confirm that the new method is faster than the leading t-SNE algorithm on large datasets, without sacrificing performance in the metrics that the authors evaluated.

**Audience:**

Yes

**Broader Impact Concerns:**

There are no broader impact concerns.

**Claims And Evidence:**

Yes

**Requested Changes:**

Critical changes:
- Compare how the final embeddings obtained by the proposed method compare to the embeddings obtained by standard t-SNE.  For example, compare the embeddings obtained by t-SNE on a dataset size of N compares to the embeddings obtained by the proposed method using an initialization size of N-1 and 1 incremental update.
- The experiments section would be greatly strengthened by repeated experiments with different random seeds so that the results tables provide error bars with the results.
- More extensive evaluation of the design choices such as the size of the initialization set and the number of gradient updates.  The former could be displayed with a curve where the x-axis is the size of the initialization set and the y-axis is a performance metric on a dataset.

**Strengths And Weaknesses:**

Strengths:
- The paper is well written and easy to read.
- There is an extensive related work section with an overview of the field.
- The presented algorithm is easy to understand and implement.

Weaknesses:
- There are design choices that are not explained theoretically or explored empirically.
    - There is no direct comparison of the original t-SNE algorithm to the the proposed algorithm to give the reader an idea of what is being traded off for improved computational efficiency.
    - It is not clear how exactly the choice of the small seed subset impacts the final performance.  While the authors do vary the initialization subset size with 3 choices (20%, 50% and 80%) in the experiments and do see that performance increases with larger initialization subsets, it is not clear how exactly this choice could affect performance because sometimes the experiments with small initialization sizes were the top performers.
   - The paper does not justify why a single gradient update is used to improve the embeddings (other than for computational reasons).  How much would the performance change if the optimization is run to completion or if a fixed number of gradient updates were used?

---

> ### Author Response · Authors · 2025-08-19
>
> We thank the Reviewer for the positive comments and suggestions.
>
> >There is no direct comparison of the original t-SNE algorithm to the proposed algorithm to give the reader an idea of what is being traded off for improved computational efficiency.
>
> We use Barnes–Hut t-SNE, a widely adopted fast variant of t-SNE. While the original version scales as $O(N^2)$, Barnes–Hut approximates it to achieve $O(N \log N)$, making it practical for large, high-dimensional datasets.
>
> >While the authors do vary the initialization subset size with 3 choices (20%, 50% and 80%) in the experiments and do see that performance increases with larger initialization subsets, it is not clear how exactly this choice could affect performance because sometimes the experiments with small initialization sizes were the top performers.
>
> Thank you for highlighting this. For example, in the iROADS dataset with DINOv2 features, the classes were clearly separable in the high-dimensional space (as can be seen from scatter plots), so even a small seed set was sufficient for accurate prototype placement and subsequent recursive updates. In such cases, adding more seed points does not significantly improve neighborhood preservation, as the global structure is already well-defined from the start.
>
> > How much would the performance change if the optimization is run to completion or if a fixed number of gradient updates were used?
>
> We added an ablation table (Table 7) evaluating different numbers of gradient steps. For example, with $\eta$ =1, a single step already achieves Silhouette ≈ 0.6858, nearly matching full convergence at ≈ 0.6912, while reducing runtime. This confirms that prototype-based initialization places new points close to high-quality positions, making extra optimization largely unnecessary.
>
> >For example, compare the embeddings obtained by t-SNE on a dataset size of N compares to the embeddings obtained by the proposed method using an initialization size of N-1 and 1 incremental update.
>
> This comparison is presented in the Results section, Subsection 4.5.3, where we evaluate the case of using $N-1$ points for initialization and one incremental update with i-RSNE against running Barnes-Hut t-SNE on all $N$ points. On DINOv2 features of the iROADS dataset, even with only one incremental update, our method still achieves slightly higher Silhouette ($0.3092$ vs $0.3088$) and better DB Index ($1.0007$ vs. $1.0070$), showing a very slight improvement even just one incremental update is applied.
>
> >The experiments section would be greatly strengthened by repeated experiments with different random seeds so that the results tables provide error bars with the results.
>
> All reported results in the tables in the revised paper include the mean and standard deviation computed over multiple runs with different random seeds.
>
> >More extensive evaluation of the design choices such as the size of the initialization set and the number of gradient updates.
>
> We have added a new ablation table analysing the effect of varying the number of gradient steps and learning rates.

---

> > ### Comment · Reviewer_Mg1i · 2025-09-04
> > **Response addresses all of my concerns**
> >
> > Thank you for the response.
> >
> > > We use Barnes–Hut t-SNE, a widely adopted fast variant of t-SNE. While the original version scales as $O(N^2)$, Barnes–Hut approximates it to achieve $O(N \log N)$, making it practical for large, high-dimensional datasets.
> >
> > I was asking about what the proposed algorithm trades off compared to the original Barnes-Hut algorithm, but my concerns seem to be addressed by the other parts of your response.
> >
> > >  in the iROADS dataset with DINOv2 features, the classes were clearly separable in the high-dimensional space ... so even a small seed set was sufficient for accurate prototype placement and subsequent recursive updates.
> >
> > Thanks, this answers my question.
> >
> > The authors responses have addressed all of my concerns.
> >
> > > We added an ablation table (Table 7) evaluating different numbers of gradient steps.
> >
> > Great, thanks for adding this!
> >
> > > This comparison is presented in the Results section, Subsection 4.5.3
> >
> > I'm not sure how I missed this, you did clearly address this.
> >
> > > All reported results in the tables in the revised paper include the mean and standard deviation
> >
> > Thanks, this is helpful.
> >
> > Your responses have addressed all of my concerns.

---

> > > ### Author Response · Authors · 2025-09-10
> > >
> > > We are glad to hear that our revisions have addressed all of your concerns, and we sincerely thank you for the constructive feedback that helped us improve the paper.

---

### Review · Reviewer_2GWp · 2025-08-04

**Summary Of Contributions:**

The authors extend the t-SNE dimension-reduction algorithm to handle online updates either in the form of an individual sample or a batch of samples. Using these modifications reduces the runtime of processing additional data while improving performance in different metrics.

**Audience:**

Yes

**Broader Impact Concerns:**

Not applicable.

**Claims And Evidence:**

Yes

**Requested Changes:**

* I think it would be helpful if you could state what the mathematical objective of t-SNE is. Then, are there any guarantees that your method is achieving the same objective? In particular, is just one gradient step sufficient?

* Isn't i-RSNE a special case of Bi-RSNE? Are there any advantages to i-RSNE when there is a small number of updates?

* If you're mathematically solving for the same objective as Barnes-Hut t-SNE, how can your performance in terms of Silhouette and Davies–Bouldin metrics be noticeably better, e.g. in Table 5? Is there an explanation/intuition for this?

**Strengths And Weaknesses:**

Please note that I'm unfamiliar with t-SNE and my review is an educated guess.

The empirical results provided seem to be quite strong. In particular, the reduction in runtime is quite significant.

On the other hand, the description of the algorithm is quite unclear to me, perhaps because I'm unfamiliar with this line of work.

---

> ### Author Response · Authors · 2025-08-19
>
> >I think it would be helpful if you could state what the mathematical objective of t-SNE is. Then, are there any guarantees that your method is achieving the same objective? In particular, is just one gradient step sufficient?
>
> The mathematical objective of t-SNE remains intact in our approach, it still minimizes the Kullback–Leibler divergence between high-dimensional similarities $p_{ij}$ and low-dimensional counterparts $q_{ij}$:
>
> $$
> \mathcal{L} = \sum_{i \neq j} p_{ij} \log \frac{p_{ij}}{q_{ij}}
> $$
>
> Here, $p_{ij}$ comes from Gaussian affinities derived from Euclidean distances in the original space, and $q_{ij}$ uses a Student-$t$ kernel on embeddings $y_i, y_j$. What changes in RSNE is the initialization: by seeding each new point near its prototype, a **single gradient-descent step** already reaches performance very close to full convergence (e.g., Silhouette 0.6530 vs 0.6442; DB Index 0.4966 vs 0.5056; from newly added Table 7 for ablations). This shows that effective initialization preserves quality while reducing runtime.
>
> >Isn't i-RSNE a special case of Bi-RSNE? Are there any advantages to i-RSNE when there is a small number of updates?
>
> The i-RSNE can be seen as the single-sample, online variant of Bi-RSNE, which processes multiple new points in a batch. The key difference lies in how similarities are computed and reused: Bi-RSNE reduce the clustering and similarity computations across many new points, while i-RSNE only updates local neighborhoods affected by the incoming point. For small numbers of updates or in real-time streaming scenarios, i-RSNE processes each new point individually; while it still updates cluster stats per insertion, it avoids recomputing summaries or distances for unaffected points, leading to lower overhead. Bi-RSNE is preferable when updates can be aggregated, as shared computations reduce the per-point cost.
>
> >If you're mathematically solving for the same objective as Barnes-Hut t-SNE, how can your performance in terms of Silhouette and Davies-Bouldin metrics be noticeably better, e.g. in Table 5? Is there an explanation/intuition for this?
>
> While both methods use a KL divergence-based objective, RSNE incorporates a prototype-guided placement step that fundamentally changes the optimization dynamics. Instead of starting from a random or PCA-based initialization, new points are placed in the embedding space using high-dimensional prototypes that preserve local neighborhood structure. This reduces the chance of early misplacement and the need for large corrective updates, leading to faster convergence and often better clustering quality. As shown in Table 5, this advantage is most visible in datasets with many well-separated classes, where meaningful prototypes provide highly accurate initial positions.

---

> > ### Comment · Reviewer_2GWp · 2025-09-06
> >
> > Thanks for your responses. I have no further questions.

---

> > > ### Author Response · Authors · 2025-09-10
> > >
> > > We appreciate your feedback and are glad that our responses have resolved your concerns.

---

### Decision · Action_Editor_Z9jb · 2025-09-12

**Recommendation:** Accept as is

**Audience:**

Yes

**Audience Explanation:**

Yes, this is an analysis of a widely use approach under a reasonable model.

**Claims And Evidence:**

Yes

**Claims Explanation:**

There were some lingering concerns from one reviewer at the time of the reviewer final decision.  However, soon after those points were responded to and a new version was uploaded.  In my view, the linger concerns have now been addressed.